# The Tip of the Iceberg: Cryopreservation Needs for Meeting the Challenge of Exceptional Plant Conservation

**DOI:** 10.3390/plants11121528

**Published:** 2022-06-07

**Authors:** Valerie C. Pence, Emily Beckman Bruns

**Affiliations:** 1Center for Conservation and Research of Endangered Wildlife (CREW), Cincinnati Zoo & Botanical Garden, 3400 Vine St., Cincinnati, OH 45220, USA; 2Department of Biological Sciences, University of Cincinnati, Cincinnati, OH 45221, USA; 3The Morton Arboretum, 4100 Illinois Route 53, Lisle, IL 60532, USA; ebeckman@mortonarb.org

**Keywords:** cryopreservation, embryo, exceptional plants, ex situ conservation, in vitro, seed

## Abstract

Cryopreservation is increasingly important as a conservation tool, particularly for threatened exceptional species. The goal of this study was to investigate the current knowledge of plant cryopreservation through a search of the literature in Web of Science and align that with the 775 species currently identified on the Working List of Exceptional Plants. While there is a good foundation in plant cryopreservation research, particularly with economically important species, there are significant gaps in research on families that contain the largest numbers of currently known exceptional species, including the Dipterocarpaceae, Rhizophoraceae, and Pittosporaceae. Even families well represented in both in the literature and on the List of Exceptional Plants had much less overlap at the level of genus. Tropical trees, a significant portion of exceptional species, were not as well represented in the literature as herbaceous species. Over 70% of all articles dealt with in vitro cryopreservation, with much less emphasis on other methods (seed, embryo, dormant bud, and pollen) that will be more cost-effective for species where they can be applied. While the research on plant cryopreservation to date provides a strong foundation and is being utilized effectively for conserving the diversity of a number of economically important species, this study revealed significant gaps that can help prioritize future research to more effectively conserve the diversity of threatened exceptional species.

## 1. Introduction

The need for cryopreservation as a conservation tool is increasing. Ex situ conservation has become an important tool for providing a back-up to the world’s plant biodiversity resources, and a network of seed banks of major crop species, including rice, wheat, corn, beans, etc., circles the globe [1,2,3]. For those crop species that either do not produce seed or are propagated clonally, a supplemental network has developed of field genebanks and more recently of cryopreserved collections, as with *Musa* (ITC, Belgium), *Solanum* (CIP, Peru), clonally propagated fruit trees, nuts (NLGRP, USA), among others [4,5,6,7].

More recently large initiatives for seed banking wild species have developed, including the Millennium Seed Bank (RBG Kew, U.K.), the Germplasm Bank of Wild Species (Kunming Institute of Botany, China), and the Australian Seed Bank Partnership (13 organizations in Australia), as well as at least 350 others in botanic gardens worldwide [8]. Such initiatives will play a critical role in meeting the ex situ conservation target of the Global Strategy for Plant Conservation of 75% of threatened species in ex situ collections [9]. However, conventional seed banks will not be able to meet this goal alone. Species with desiccation sensitive (recalcitrant) seeds, for example, cannot undergo the drying required for seed banking, and a recent study indicated that such species are projected to make up more than a quarter of threatened species [10]. These and other species, such as those without adequate seeds for banking or seeds that are short-lived in storage, pose challenges to conventional seed banking and are known as exceptional species [11]. 

As with clonally propagated crop species, cryopreservation is being explored for its use in maintaining threatened exceptional species in ex situ collections [12,13]. However, unlike crop species, where a protocol developed for a particular crop is adapted to many accessions, the numbers and diversity of wild threatened species will require much more research into adapting cryopreservation protocols to species with a wide variety of growth habits and adaptations. There have been several programs that have focused on wild species cryopreservation (e.g., Kings Park and Botanic Gardens/Australian PlantBank (Australia), the Cincinnati Zoo & Botanical Garden’s CREW CryoBioBank (US), etc.), but much more capacity is needed. There are predicted to be more than 24,000 exceptional species worldwide [14,15]. With the estimation that 40% of plant species are threatened [16], the number of exceptional species that could benefit from cryopreservation efforts is projected to be more than 10,000. However, a recent analysis indicated that cryopreservation is underutilized as a tool for conserving exceptional plants [17], with only 1% of identified exceptional species reported as being held in cryo-collections. There is, thus, a critical need for efforts to increase its use.

As a first step in understanding these needs more specifically, an analysis of the current literature available on the cryopreservation of exceptional plants was made. Although the identity of all the species projected to be exceptional is not known, this analysis used a recently published Global Working List of Exceptional Plants, consisting of 775 identified exceptional species [17] as a starting point. That list categorizes exceptional species by the reason for their exceptionality, known as a species’ exceptionality factor, or EF, which can help identify the conservation approaches that will be the most efficient and capture the most genetic diversity for that species (Figure 1). Most of these approaches will need some form of cryopreservation, and this study worked to align the needs of exceptional plants with the current literature on cryopreservation to identify strengths and gaps in knowledge and to help set priorities for future areas of research.

## 2. Results

A total of 1,798 articles were identified from Web of Science that involved some aspect of seed-bearing plant seed or tissue cryopreservation representing 128 families and 490 genera. These were compared with the 111 families and 366 genera on the Working List of Exceptional Plants (hereafter referred to as “exceptional families” and “exceptional genera” for convenience, although not all species in these families or genera are necessarily exceptional). There were 74 families in common between the two datasets, with 37 families on the List of Exceptional Plants not represented in the literature. The four families with the most cryopreservation articles were Rosaceae, Orchidaceae, Solanaceae, and Poaceae, while the four exceptional families with the most identified species were Dipterocarpaceae, Arecaceae, Rutaceae, and Campanulaceae (Table 1).

When the 25 families with the most cryopreservation articles were compared with the 25 exceptional families with the most species, there were 13 families in common among the top 25 and four in common among the top 10: Orchidaceae, Arecaceae, Rutaceae, and Fabaceae. Three of the top 25 exceptional families were not represented in the literature at all: Dipterocarpaceae, Rhizophoraceae, and Pittosporaceae, while there were two exceptional families that had more than 100 articles each: Orchidaceae and Poaceae.

When genera were similarly examined, there were 92 genera in common between the two datasets and 274 genera on the List of Exceptional Plants not represented in the literature search. The top three genera with the most cryopreservation articles were *Solanum, Malus*, and *Citrus*, while the top three exceptional genera were *Shorea, Cyanea*, and *Quercus* (Table 2). Only three genera were in common among the top 25 genera from the literature and the top 23 exceptional genera: *Citrus*, *Coffea*, and *Quercus. Citrus* had the third highest number of articles of any genus, 62, while *Coffea* and *Quercus* had 28 and 23, respectively. There were 11 exceptional genera in the top 23 with no representation in the cryopreservation literature, including the two with the most identified exceptional species, *Shorea* and *Cyanea*.

When the four families in common in the top 10 of both groups were examined at the level of genus, exceptional genera that were also found in the cryopreservation literature ranged from 17% of exceptional Rutaceae genera to 52% of exceptional Orchidaceae genera (Table 3). The one genus in common in the Rutaceae was *Citrus*. The five Fabaceae genera in common were *Astragalus*, *Crotalaria*, *Inga*, *Vicia*, and *Vigna*. Of these five, only *Inga* was found to have a high percentage of tree species, as determined by GlobalTreeSearch (Table 4). The other four are known to be primarily herbaceous, although also containing some shrubs. In contrast, of the remaining 19 exceptional Fabaceae genera, 15 had more than 50% of their species classified as trees, with many being tropical genera. The two remaining genera with no trees, *Kanaloa* and *Strongylodon*, include shrubs or woody vines.

When the 775 species on the current List of Exceptional Plants were cross-referenced with the literature search, there were 18 of the 775 species with five or more articles in the literature search, with *Carica papaya* and *Cocos nucifera* having more than 20 articles each (Table 5). Most of these species are well-known as food crops or of other economic importance and 11 of the 18 had more than 20 other exceptional species identified in the same family. In the cases of *Carica papaya* (Caricaceae), *Persea americana* (Lauraceae), *Mangifera indica* (Anacardiaceae), and *Diospyros kaaki* (Ebenaceae), half or more of the articles in the family were of that one species.

There were 19 exceptional families for which no articles were found in the literature search, representing 152 species, or 20% of the List of Exceptional Plants (Table 6). When these were examined for the reason for their exceptionality (their exceptionality factor or EF; [11]), the highest proportion of these were EF2 (desiccation sensitive/recalcitrant) (70%), followed by EF3 (freeze-sensitive/short-lived) (26%). Dipterocarpaceae was the family with the most exceptional species (59) with no literature in the Web of Science search, and 55 or 93% of these were classified as EF2. To examine the lack of literature more fully, a separate search for “Dipterocarpaceae and cryopreservation” was made using Google Scholar and reviewing the first four pages of results. This produced nine results (Table 7), including references to conference and workshop proceedings, a poster presentation, institutional reports, and two articles in journals not included in Web of Science, dealing with a total of 15 dipterocarp species [20,21,22,23,24,25,26,27,28]. In contrast, a similar search of Google Scholar for “Rhizophoraceae and cryopreservation,” the family with the next highest number of exceptional species and no Web of Science literature, produced no references relevant to cryopreservation. Searching for “Pittosporaceae and cryopreservation” yielded one article on the cryopreservation of seeds of multiple species that were not identified in the title or abstract, and thus had been excluded from the Web of Science search, but which included information on freezing seeds of one Pittosporaceae species [29].

When the Web of Science articles on plant cryopreservation were categorized with regard to the type of tissue used in the study, over 80% of the articles involved in vitro tissues (Figure 2). There were 351 genera represented in literature on the cryopreservation of in vitro tissues, 131 for seeds, 87 for embryos, 51 for pollen, and 19 for dormant bud studies. When the top 10 genera were listed by the type of tissue cryopreserved, many genera were represented by research in the top ten of more than one type of tissue (Table 8).

This literature search also captured cryopreservation articles on non-seed plants, algae, and fungi, many of which will also require cryopreservation for long-term, effective cryopreservation. The largest number of articles dealt with algae cryopreservation (27), followed by fungi (22), pteridophytes (15), and bryophytes (13) (Figure 3). These included eight genera of bryophytes representing seven families and nine genera of pteridophytes representing seven families (Table 9).

## 3. Discussion

Cryopreservation will be an essential tool for the ex situ conservation of tissues from exceptional plants [30]. This study aligned almost 1800 articles dealing with plant cryopreservation downloaded from Web of Science with the List of Exceptional Plants (775 species) [17] to investigate areas where cryopreservation research has been concentrated and to identify gaps where work on exceptional plants should be prioritized. The Working List of Exceptional Plants resulted from an evaluation of exceptional status of over 23,000 species and is available online (https://cincinnatizoo.org/global-list-of-threatened-exceptional-plants/, accessed 31 May 2022). The Web of Science download was based on searches using keywords relevant to plant cryopreservation that yielded focused results that could be refined in a semi-automated way. While the Web of Science coverage of the literature is extensive, it does not include some sources, such as lower impact journals and gray literature, as demonstrated by the search of Google Scholar for the Dipterocarpaceae. However, a similar search for two other exceptional families with no Web of Science literature yielded information on only one species, suggesting that the Web of Science was providing a reasonable overview of the current state of plant cryopreservation. Web of Science also provided full title and abstract text for download that could be adapted to a partially automated methodology. The iterative search methodology used in this study to avoid non-target genera could also miss target genera listed only in the abstract if a different target genus was listed in the title, although from our manual reviews this appeared to be uncommon.

The resulting information was examined primarily at the family and genus level. The literature search revealed that the coverage of plant cryopreservation studies has been fairly broad, covering 490 genera in 128 families. Not surprisingly, much of it has been focused on families that are of economic importance, particularly those of major food, timber, and horticultural crops, such as Poaceae (e.g., *Oryza, Saccharum*), Solanaceae (e.g., *Solanum*), Rutaceae (e.g., *Citrus*), Rosaceae (e.g., *Malus, Pyrus, Prunus, Rubus, Fragaria*), and others. In comparing this search with the List of Exceptional Plants there were 74 families in common, or 67% of the exceptional families. The Orchidaceae and the Poaceae, both in the top 25 families with exceptional species, had over 100 articles each, likely reflecting the commercial importance of these two families, and providing a good basis for work with additional taxa in these groups [31,32]. However, comparing the two lists at the level of genus revealed only 92 genera in common, representing only 25% of the currently known exceptional genera. This difference in emphasis between research as reflected in the literature and exceptional plant taxa was further seen in the low proportion of genera in common within the four families that were in the top 10 of both lists, ranging from 17% in the Rutaceae (representing the single genus *Citrus*) to 52% (representing 10 common genera) in the Orchidaceae, a large family that has had particular attention both commercially and for conservation [33,34,35]. Cryopreservation research has been focused largely on species of economic importance and there is a need to widen the scope of investigation to wild exceptional species, particularly threatened taxa. As a number of these are confamilial with crop species, there is a foundation of work for this effort.

Many of the exceptional genera not represented in the literature are woody species, particularly tropical woody taxa. The three families in the top 25 exceptional families that were not found in the cryopreservation literature search—Dipterocarpaceae, Rhizophoraceae, and Pittosporaceae—are all families of tropical woody species with significant ecological, medicinal, and/or economic importance [36,37,38]. Within the Fabaceae, a family that was in the top 10 of both lists, four of the 5 genera in common are primarily herbaceous species with the majority of the exceptional Fabaceae being tropical woody species. Predictive models have indicated that a high proportion of exceptional species will be woody species from the tropics [14,39] and many of these are already represented on the List of Exceptional Plants. With the level of threat to species in the tropics increasing [40,41,42], tropical trees should be prioritized for cryopreservation research.

Several types of cryopreservation research will be needed for conserving exceptional plants, depending on the type of tissue available, and these will include cryopreserving whole seeds, isolated zygotic embryos, dormant buds, or in vitro tissues (shoot tips or somatic embryos), as well as pollen. This study indicated that, by far, the largest emphasis in the cryopreservation literature has been on cryopreserving in vitro tissues, with 73% of all the plant cryopreservation articles focused on this approach. This largely reflects organized programs of in vitro tissue cryopreservation for banking commercially important genera and the research that has formed the basis for these efforts. Of the 10 genera with the most articles on in vitro cryopreservation, at least six are being systematically cryobanked using in vitro tissues (*Solanum, Malus, Allium, Vitis, Musa, Prunus*) [5,6,7,43,44,45]. This is largely to maintain valuable clonal lines, since many species in these genera would not be classified as exceptional (i.e., their seeds can be maintained in conventional seed banks). Cryopreservation of in vitro tissues is the most labor and resource intensive of all the approaches for plant cryopreservation, as it requires both in vitro methods for generating shoot tips or somatic embryos for cryopreservation, as well as for growing the tissues after cryostorage and recovering plants. However, work with these crop species has demonstrated the feasibility of banking in vitro tissues on a larger scale, and this work can inform efforts to conserve threatened exceptional species.

However, if other, less labor and resource intensive methods for cryopreserving are workable for an exceptional species, they should be prioritized. A large number of exceptional species are predicted to be short-lived seeds, i.e., seeds that are somewhat desiccation tolerant but short-lived at conventional seed banking temperatures [15]. In many cases, simply substituting cryopreservation for conventional temperatures of seed bank freezers (−20 °C) can extend longevity significantly [46] and would be the most efficient method of long-term conservation for these species. However, seed cryopreservation was the focus of only 11% of the articles in this study. Given that short-lived seed species will likely number in the thousands [15], this method should receive more attention.

Two other cryopreservation methods that are also generally more cost-effective than cryopreserving in vitro tissues are freezing isolated embryos and dormant buds. Neither of these methods requires generating an in vitro culture line before cryopreservation, although in vitro or other methods are needed for recovery after cryo. Embryo cryopreservation has been demonstrated for a number of short-lived and recalcitrant species, particularly those with larger seeds, such as *Quercus robur* and *Juglans nigra* [47,48]. Dormant bud freezing has been successful for several cold-hardy woody species and has been adopted as a method for long-term banking for collections, including those of *Malus*, *Morus*, and *Ulmus* [49,50,51,52]. Despite their potential, only 9% and 2% of the cryopreservation literature dealt with embryo or dormant bud cryopreservation, respectively. These methods might be particularly adaptable for recalcitrant species, which currently comprise 50% of the List of Exceptional Plants [17] and should be the focus of increased research.

Pollen cryopreservation can be a valuable supplemental method for conserving plant genetic diversity, allowing crosses across geographic and temporal distances that can help maintain and support the health of threatened populations. The methods are similar to seed banking, requiring some drying before freezing and have been shown to be workable for many species [53]. Despite its relative simplicity compared with other cryopreservation methods, pollen cryopreservation was the subject of only 4% of the literature in our search. 

While our study focused on seed plants, a number of articles were captured dealing with non-seed plants and fungi, although a targeted search for these groups might reveal more. Cryopreservation of spores of pteridophytes has been successful with a number of species [54,55,56], and it is likely that many fern species will require cryopreservation for spore conservation [57]. Gametophytes of both bryophytes and pteridophytes have also been successfully cryopreserved [56,58,59], providing another tool that could be useful for rare species producing few spores, and both methods should be more widely applied for the conservation of rare ferns and bryophytes.

The goal of securing 75% of the world’s threatened species in ex situ conservation, set by the Global Strategy for Plant Conservation [9], will not be achieved without workable and cost-effective methods of cryopreservation of seeds and tissues of a wide variety of species from diverse habitats. While our analysis of the plant cryopreservation literature at the level of genus could not be aligned directly with threat status, other studies have shown that more than a quarter of identified exceptional species are known to be threatened, and these will be numbered in the thousands [10,17]. To meet this challenge, the question will be, can cryobanking of wild threatened species be scaled up in a way that mirrors the major seed banking efforts? 

The current cryopreservation literature provides a foundation for approaching this goal, and there are several large-scale cryo-banking efforts of clonally propagated crops and high value varieties of fruit and nut trees that can provide models for banking exceptional species on a larger scale [6,44,60]. However, there will be some differences in utilizing cryopreservation more broadly for conservation compared with banking varieties of crop species. Although genotypic variation within species has been encountered in both domestic and wild species [61,62], developing methods for a diverse range of wild taxa with very different adaptations will be even more challenging. Building on and diversifying the plant cryopreservation literature is needed to provide new knowledge and insights to advance the science and technologies needed for conservation. This literature search has highlighted exceptional genera and families not represented in past research that should be prioritized for cryopreservation work moving forward. As cryopreservation research is expanded to a wider range of species, it should be possible to analyze the methods and results for patterns that can be used to predict and streamline protocols for new species. Widening the scope of species studied to develop this “comparative cryobiotechnology” will be necessary to develop the much needed, large-scale programs to meet the challenges of conserving exceptional plants.

## 4. Materials and Methods

Using Biosis Previews of Web of Science (Clarivate Analytics) accessed through the University of Cincinnati Libraries, a search was made on 11 March 2022 for “cryopreservation + plant”, which yielded 2424 results. On March 15, a search was made for “cryopreservation + seed”, which yielded 1277 results. On April 3, a search for “liquid nitrogen + plant” was also attempted (5417 results) but most of these results did not deal with plant cryopreservation, so this search was not included in the study. Results from the “cryopreservation + plant” and “cryopreservation + seed” searches were downloaded from the Web of Science as Excel files.

All Excel files were imported into R for analysis [63] and duplicate articles were removed using the Web of Science unique ID; this resulted in 3015 articles for analysis. Next, the World Flora Online (WFO) Taxonomic Backbone [19] was used to create a list of all unique genera and a list of all unique families. Using the rebus R package [64], a search for each genus and each family in the WFO lists was made in both the title and abstract of all downloaded Web of Science articles. A few genera were excluded from the word search because they only matched instances of the words that did not refer to the genus; these included “Aa”, “Cotyledon”, “Cuba”, “India”, “Ion”, and “Medium.” Because some articles use only a species’ common name, a list of frequently-used common names—mostly economically-important species—and their corresponding scientific names was manually constructed.

All genus and family names were searched for iteratively in the title and abstract to avoid selecting non-target names (e.g., genera/families mentioned in the title/abstract but that were not the subject of the research) as much as possible. Each iteration included one pass of the articles while searching for genus names and one pass while searching for family names, with articles being removed from the next iteration as soon as a name(s) was matched. The iterations went as follows: (1) search the title for WFO genus/family names, (2) search the title for common names using our manually-compiled list, (3) search the abstract for WFO genus/family names, (4) search the abstract for common names. Searching iteratively often avoided non-target genera being found, but it also sometimes resulted in target genera being missed. For example, if an article’s title listed one genus and the abstract listed additional target genera, only the genus in the title would be recorded in our analysis. Of the 3015 articles searched, 735 had no genus match and 2758 had no family match. For articles with a genus/genera found and no family, the genus/genera were used to determine the family/families. This resulted in 710 articles with no family identified. Finally, genus and family names were manually spot-checked and edited for accuracy.

To further confirm that the articles dealt with cryopreservation of plants, article titles and abstracts were also searched for lists of manually-curated keywords. Keywords included both ‘positive’ words that signaled the article was likely of interest (e.g., cryopreserve, cryogenic, liquid nitrogen) and ‘negative’ keywords that indicated the article probably wasn’t of interest either because it was not focused on plants or was a review rather than original findings (e.g., mammal, sperm, review). ‘Non-seed’ keywords were also searched for, to identify if the article was likely related to non-seed plants (e.g., algae, moss, fern). Articles that had negative keywords were removed from the analysis, except those with the word “human”, which were manually reviewed to determine if they were of interest or not; articles without positive keywords were spot-checked to look for additional keywords that could be included; articles with non-seed keywords were manually reviewed to check this categorization. Additional articles falling into each category were also manually identified during this process. Finally, keywords indicating the type of cryopreservation reported in the article (e.g., dormant bud, in vitro, zygotic embryo) were searched in the title and abstract and, based on these keywords, each article was categorized as Dormant Bud, Embryo, In Vitro, Pollen, and/or Seed.

## Figures and Tables

**Figure 1 plants-11-01528-f001:**
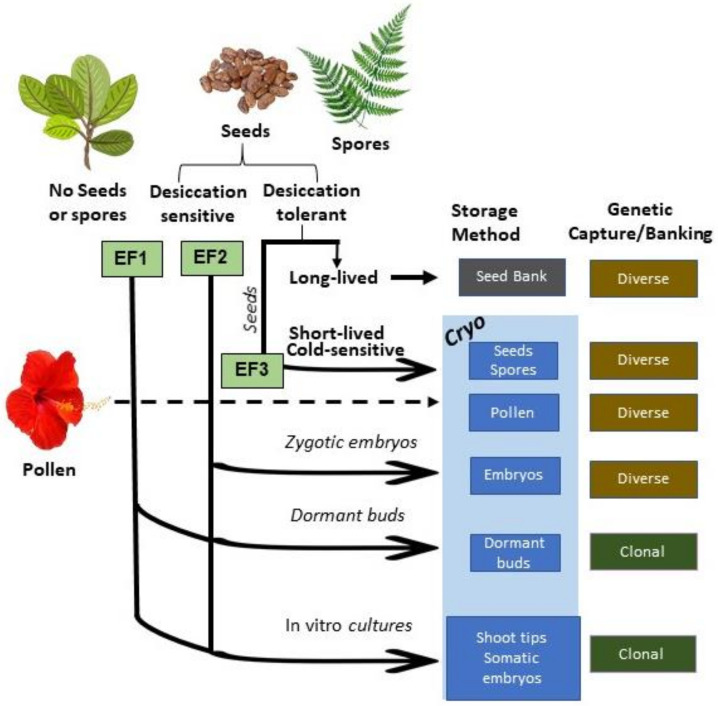
The role of cryopreservation in conserving plant seeds and tissues ex situ and relationship to exceptionality factor (EF) and genetic capture. EF1 = seeds unavailable; EF2 = seeds desiccation sensitive; EF3 = seeds short-lived.

**Figure 2 plants-11-01528-f002:**
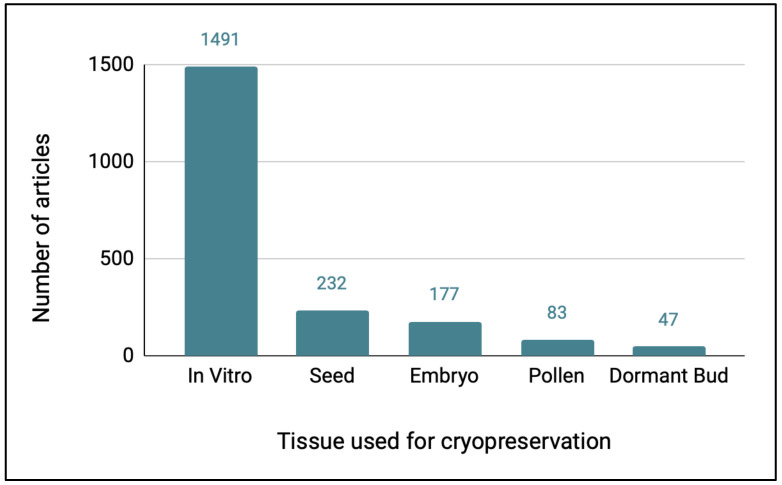
Number of Web of Science articles identified in each of the five categories of plant cryopreservation research, separated by the type of tissue used in the study.

**Figure 3 plants-11-01528-f003:**
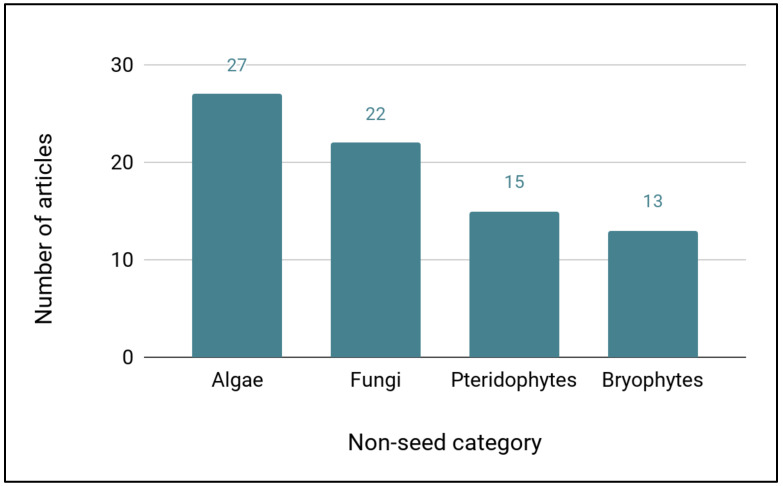
Number of Web of Science articles identified for each category of non-seed plants, algae, and fungi.

**Table 1 plants-11-01528-t001:** The top 25 families with the most Web of Science articles identified, compared with the 25 families on the Working List of Exceptional Plants with the largest number of species currently known to be exceptional.

Literature Search		Exceptional Plant List		
Family	No. of Articles	Family	No. of Species	No. of Articles
Rosaceae	183	Dipterocarpaceae	59	0
Orchidaceae *	134	Arecaceae *	37	71
Solanaceae	106	Rutaceae *	35	69
Poaceae	105	Campanulaceae	34	1
Pinaceae	80	Fabaceae *	32	66
Asteraceae	74	Rubiaceae	32	35
Arecaceae *	71	Orchidaceae *	26	134
Rutaceae *	69	Meliaceae	25	30
Amaryllidaceae	67	Fagaceae	24	37
Fabaceae *	66	Lauraceae	24	9
Brassicaceae	49	Sapindaceae	23	21
Lamiaceae	45	Asteraceae	22	74
Vitaceae	40	Primulaceae	20	8
Fagaceae	37	Amaryllidaceae	18	67
Rubiaceae	35	Moraceae	18	20
Musaceae	33	Myrtaceae	18	18
Euphorbiaceae	30	Rhizophoraceae	15	0
Meliaceae	30	Apocynaceae	14	24
Bromeliaceae	29	Malvaceae	12	29
Malvaceae	29	Sapotaceae	12	2
Dioscoreaceae	27	Poaceae	11	105
Salicaceae	25	Anacardiaceae	10	14
Apocynaceae	24	Araucariaceae	10	6
Betulaceae	22	Gesneriaceae	10	3
Sapindaceae	21	Pittosporaceae	10	0

Shaded families are those in common in the top 25 of each list, while starred (*) families are those in common in the top 10 of each list.

**Table 2 plants-11-01528-t002:** The top 25 genera with the most Web of Science articles identified, compared with the 23 genera with six or more species listed in the current Working List of Exceptional Plants.

Literature Search		Exceptional Plant List		
Genus	No. of Articles	Genus	No. of Species	No. of Articles
*Solanum*	80	*Shorea*	26	0
*Malus*	73	*Cyanea*	20	0
*Citrus **	62	*Quercus*	17	23
*Allium*	49	*Artocarpus*	15	7
*Prunus*	46	*Melicope*	15	0
*Dendrobium*	42	*Coprosma*	12	0
*Vitis*	40	*Lysimachia*	12	0
*Picea*	35	*Dipterocarpus*	11	0
*Pinus*	35	*Citrus **	10	62
*Musa*	32	*Cyrtandra*	10	0
*Coffea*	28	*Pittosporum*	10	0
*Dioscorea*	27	*Araucaria*	9	5
*Arabidopsis*	26	*Hopea*	9	0
*Oryza*	26	*Inga*	9	1
*Chrysanthemum*	25	*Syzygium*	9	5
*Pyrus*	25	*Clermontia*	8	0
*Phoenix*	24	*Aesculus*	7	5
*Fragaria*	23	*Garcinia*	7	8
*Quercus*	23	*Bruguiera*	6	0
*Rubus*	22	*Coffea*	6	28
*Mentha*	21	*Diospyros*	6	9
*Saccharum*	20	*Rhizophora*	6	0
*Hypericum*	17	*Trichilia*	6	11
*Lilium*	17			
*Populus*	17			

There are eight genera with five species on the exceptional plant list, therefore these are not shown in the table. Shaded genera are those in common between the two lists, while starred (*) genera are those in common in the top 10 of each list.

**Table 3 plants-11-01528-t003:** Genera in common between the Exceptional Plant List and the Web of Science literature search, for the three families in both the top 10 families on the Exceptional Plant List and the top 10 families with the most articles from the literature search.

Family	Exceptional Species Genera	Literature Search Genera	No. in Common	Percent Exceptional Species Genera in Literature
Arecaceae	23	14	7	30%
Fabaceae	24	36	5	21%
Orchidaceae	19	44	10	52%
Rutaceae	6	5	1	17%

**Table 4 plants-11-01528-t004:** Genera of Fabaceae with exceptional species evaluated for the percent of species of trees within each genus, as determined from GlobalTreeSearch [18].

Fabaceae with Exceptional spp.	GlobalTreeSearch Number of spp. (Trees)	WFO Synonym spp.	WFO Accepted spp.	WFO Unchecked spp.	WFO Doubtful spp.	Approximate Percent Trees ^a^
*Andira*	29	31	40	1	NA	71%
*Astragalus*	0	2308	3108	146	23	0%
*Castanospermum*	1	2	1	NA	NA	100%
*Cojoba*	14	20	15	2	NA	82%
*Copaifera*	34	38	45	4	1	69%
*Cordyla*	5	5	5	1	NA	83%
*Crotalaria*	6	596	716	16	24	1%
*Cynometra*	108	72	88	4	NA	117%
*Detarium*	3	4	3	NA	1	100%
*Dipteryx*	11	16	12	2	NA	79%
*Erythrina*	107	152	132	16	10	72%
*Inga*	266	504	279	28	13	87%
*Kanaloa*	0	NA	1	NA	NA	0%
*Marmaroxylon*	0	9	NA	NA	NA	* See footnote
*Pentaclethra*	3	7	3	1	NA	75%
*Prioria*	11	NA	14	NA	NA	79%
*Saraca*	10	23	11	1	NA	83%
*Senna*	110	36	282	20	NA	36%
*Sesbania*	13	57	63	11	NA	18%
*Sophora*	29	121	63	16	NA	37%
*Strongylodon*	0	12	16	3	NA	0%
*Swartzia*	185	79	199	14	NA	87%
*Vicia*	0	404	248	86	2	0%
*Vigna*	0	202	104	15	NA	0%
*Zygia*	55	38	66	4	NA	79%

* See data for *Zygia*, which is sometimes considered a synonym of *Marmaroxylon*. **^a^** Not exact species match; calculated using number of WFO Accepted and Unchecked spp. Green = genera in common between Web of Science search and exceptional list. Purple = approximate percent of trees >50%. WFO = World Flora Online [19].

**Table 5 plants-11-01528-t005:** Exceptional species listed in the current Working List of Exceptional Plants, which were found in five or more articles in the Web of Science literature search.

Exceptional Species	No. of Articles	Family	No. of Identified Exceptional Species in Family	No. of Total Articles in Family	Percent of Total Articles on One Species
*Carica papaya*	20	Caricaceae	1	20	100%
*Cocos nucifera*	21	Arecaceae	37	71	30%
*Coffea arabica*	17	Rubiaceae	32	35	49%
*Citrus sinensis*	14	Rutaceae	35	69	20%
*Elaeis guineensis*	12	Arecaceae	37	71	17%
*Mangifera indica*	11	Anacardiaceae	10	14	79%
*Theobroma cacao*	11	Malvaceae	12	29	38%
*Trichilia dregeana*	10	Meliaceae	25	30	33%
*Artocarpus heterophyllus*	9	Moraceae	18	20	45%
*Castanea sativa*	8	Fagaceae	24	37	22%
*Quercus robur*	8	Fagaceae	24	37	22%
*Diospyros kaki*	7	Ebenaceae	6	9	78%
*Persea americana*	7	Lauraceae	24	9	78%
*Citrus limon*	6	Rutaceae	35	69	9%
*Ekebergia capensis*	6	Meliaceae	25	30	20%
*Hevea brasiliensis*	6	Euphorbiaceae	2	30	20%
*Passiflora edulis*	5	Passifloraceae	1	14	36%
*Wasabia japonica*	5	Brassicaceae	2	49	10%

**Table 6 plants-11-01528-t006:** Families with two or more species listed in the current Working List of Exceptional Plants, but with no articles identified in the Web of Science literature search.

Family	Number of Exceptional Species				
	Total	EF1	EF2	EF3	EF4
Dipterocarpaceae	59	0	55	4	0
Rhizophoraceae	15	0	15	0	0
Pittosporaceae	10	1	0	10	0
Myristicaceae	9	0	7	2	0
Podocarpaceae	7	0	5	2	0
Santalaceae	7	1	1	5	0
Nymphaeaceae	6	0	4	1	1
Cyperaceae	5	0	0	5	1
Nyctaginaceae	5	0	4	1	0
Urticaceae	5	0	0	5	1
Cymodoceaceae	4	0	4	0	0
Lecythidaceae	4	0	4	0	0
Dilleniaceae	3	0	0	0	3
Scrophulariaceae	3	0	0	3	0
Calophyllaceae	2	0	2	0	0
Chrysobalanaceae	2	0	2	0	0
Elaeocarpaceae	2	1	1	0	0
Hydrangeaceae	2	0	0	2	0
Zosteraceae	2	0	2	0	0
TOTAL	152	3	106	40	6
PERCENT		2%	70%	26%	4%

Exceptionality factors (EFs) of the species within each family are also provided. EF1 = few or no seeds; EF2 = desiccation sensitive seeds; EF3 = short-lived seeds; EF4 = seeds deeply dormant [11]. Shaded cells indicate non-zero values.

**Table 7 plants-11-01528-t007:** Articles found with Google Scholar for “Dipterocarpaceae and Cryopreservation”.

Type of Publication	Reference
Journal articles: *Journal of Tropical and Subtropical Botany* [Chinese]; *International Journal of Agriculture for Plantations*	[20,21]
Conference proceedings	[22,23]
Workshop proceedings	[24]
Symposium proceedings	[25]
Poster presentation cited in a report	[26]
Institutional reports	[27,28]

Dipterocarpaceae was the family with the most exceptional species that had no cryopreservation articles found with the Web of Science search.

**Table 8 plants-11-01528-t008:** Top 25 genera with the most Web of Science articles in each of the five categories of plant cryopreservation research.

In Vitro		Embryo		Seed		Pollen		Dormant Bud	
Genus	No. of Articles	Genus	No. of Articles	Genus	No. of Articles	Genus	No. of Articles	Genus	No. of Articles
*Solanum*	70	*Coffea*	13	*Citrus **	11	*Paeonia*	7	*Malus*	19
*Malus*	60	*Citrus **	11	*Cocos*	9	*Solanum*	6	*Morus*	3
*Allium*	42	*Prunus **	8	*Trichilia*	8	*Allium*	4	*Prunus **	3
*Citrus **	39	*Passiflora*	6	*Coffea*	7	*Brassica*	3	*Ribes*	3
*Dendrobium*	39	*Pinus*	5	*Quercus*	6	*Citrus **	3	*Diospyros*	2
*Vitis*	37	*Salix*	5	*Camellia*	5	*Dendrobium*	3	*Populus*	2
*Picea*	28	*Allium*	4	*Livistona*	5	*Rosa*	3	*Pyrus*	2
*Pinus*	28	*Camellia*	4	*Acer*	4	*Camellia*	2	*Salix*	2
*Musa*	27	*Malus*	4	*Castanea*	4	*Carya*	2	*Actinidia*	1
*Prunus **	27	*Musa*	4	*Elaeis*	4	*Elaeis*	2	*Citrus **	1
*Dioscorea*	25	*Picea*	4	*Ilex*	4	*Lilium*	2	*Eucalyptus*	1
*Arabidopsis*	24	*Populus*	4	*Musa*	4	*Olea*	2	*Fraxinus*	1
*Oryza*	24	*Pyrus*	4	*Picea*	4	*Phoenix*	2	*Juglans*	1
*Chrysanthemum*	22			*Prunus **	4	*Prunus **	2	*Petunia*	1
*Fragaria*	20			*Amaryllis*	3	*Ricinus*	2	*Phoenix*	1
*Phoenix*	19			*Jatropha*	3			*Quercus*	1
*Saccharum*	19			*Zea*	3			*Rosa*	1
*Mentha*	18							*Ulmus*	1
*Rubus*	18							*Vaccinium*	1
*Hypericum*	17								

Shading indicates genera that are only in the top 25 within one category, while starred (*) genera are those in common in the top 25 across all five categories.

**Table 9 plants-11-01528-t009:** Genera and families of bryophytes and pteridophytes captured in this literature search, representing cryopreservation work on these taxa.

Bryophytes		Pteridophytes	
Family	Genus	Family	Genus
Marchantiaceae	*Marchantia*	Osmundaceae	*Osmunda*
Ditrichaceae	*Ditrichum*	Pteridaceae	*Ceratopteris*
Funariaceae	*Physcomitrella*	Aspleniaceae	*Asplenium*
Bryaceae	*Bryum*	Polypodiaceae	*Pleopeltis*
Pilotrichaceae	*Cyclodictyon*	Cyatheaceae	*Cyathea*
Splachnaceae	*Splachnum*	Salviniaceae	*Azolla*
Polytrichaceae	*Pogonatum*	Equisetaceae	*Equisetum*
Polytrichaceae	*Polytrichum*	Cyatheaceae	*Alsophila*
		Aspleniaceae	*Neottopteris*

## Data Availability

Restrictions apply to the availability of these data. Data was obtained from Web of Science and are available from the authors with the permission of Web of Science. Supporting data and scripts used for analysis are publicly available at https://github.com/esbeckman/WoS_wordsearch_cryopres.

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
