# Peer review of "The Tip of the Iceberg: Cryopreservation Needs for Meeting the Challenge of Exceptional Plant Conservation"

_plants, 2022, doi:10.3390/plants11121528_

Round 1

Reviewer 1 Report

The Tip of the Iceberg: Cryopreservation Needs for Meeting the Challenge of Exceptional Plant Conservation

This is a well written review that comprehensively covers the current use of cryopreservation for the conservation of exceptional species, comparing the 775 listed Global Working List of Exceptional Plants to the published literature on plant cryopreservation. This review identifies the Families and Genera where future research is needed, as well as highlighting the limited work that has been done utilising cryopreservation for alternative germplasm sources such as seed, pollen and dormant buds. This provides a clear foundation for where the authors suggest future cryopreservation research should be conducted to best conserve threatened exceptional plants.

Some small minor corrections are listed below

Line 65 - “Fitableg” correct to Fig

Table 2 – It is not clear why the authors have listed 25 genera from the Web of Science search but only 23 genera from the Exceptional Plant List. The table just looks a little incomplete with the 2 missing rows of data.

Table 3 - include the percent symbol in the last columns data, to be consistent with the other tables where percent is used

Table 4 – include full names for the WOS and WFO abbreviations used

Table 9 – Suggest swapping the columns so that Family is in the first column and Genus the second column, with the data sorted alphabetically. This table also has an empty column separating the Bryophytes from Pteridophytes that can be deleted

Line 281 – that the correct degrees symbol?

There is a large number of tables, perhaps some of the tables could be linked as supplementary information, such as table 7 and table 9

Reviewer 2 Report

The genera and family level survey of plant cryopreservation and comparison with exceptional plant list attracted me. Some issues need to be addressed are as follow:

1. The term "exceptional" seems less scientific to me, since it's natural in nature. I understand the term represents non-orthodox seeds and vegetatively propagated species defined by FAO Genebank standards. Though the concept of EF1, EF2, EF3 looks fine, I cannot find the definition for EF4 (Table 6).

2. In terms of selecting plant materials for the cryopreservation of wild species (especially threatened), the priority is the availability of the samples in quality and quality, as well as the storage characteristics of materials. 

3. In depth analysis between genera/family and tissues is highly welcomed by the readers. 
